# Microenvironmental Gradients Drive Spatial Stratification of Saccharifying Microbial Communities and Enzyme Activity in Strong-Flavor *Daqu* Fermentation

**DOI:** 10.3390/foods14234160

**Published:** 2025-12-04

**Authors:** Wenyi Jiang, Suyi Zhang, Zhiping Feng, Yi Dong, Zonghua Ao, Junjie Jia, He Li, Zhilin Chen, Ruidi Liu, Xingke Wen

**Affiliations:** 1College of Bioengineering, Sichuan University of Science & Engineering, Yibin 644005, China; 322086002311@stu.suse.edu.cn; 2Luzhou Laojiao Co., Ltd., Luzhou 646000, China; 3National Engineering Research Center of Solid-State Manufacturing, Luzhou 646000, China

**Keywords:** strong-flavor *Daqu*, saccharifying enzymes, spatiotemporal dynamics, metagenomics, microbial communities

## Abstract

*Daqu*, a representative solid-state fermentation product, produces saccharifying enzymes to degrade sorghum starch into fermentable sugars for ethanol synthesis. Spatial heterogeneity in *Daqu* drives community assembly. However, its regulatory role in enzyme-driven saccharification remains unclear. By integrating metagenomics and PacBio full-length sequencing, this study investigated how microenvironmental gradients across distinct *Daqu* layers (QP (surface layer), HQ (middle layer), QX (center layer)) shape saccharifying microbiota and activity. Saccharifying activity exhibited a declining surface-to-center gradient (e.g., QP: 870.9 ± 21.2 U/mL > HQ: 631.2 ± 16.4 U/mL > QX: 296.5 ± 16.1 U/mL on day 30, *p* < 0.05), paralleled by divergence in microenvironments. Metagenomics identified α-amylase and α-glucosidase as key saccharifying enzymes, primarily encoded by fungi; their abundance was inhibited by heat and humidity, yet promoted by acidity. Enzymatic validation confirmed higher saccharifying activity in QP and HQ core microbes (e.g., *Lichtheimia ramosa*: 43.16 ± 1.97 U/mL) than in QX (e.g., *Paecilomyces variotii*: 14.27 ± 1.25 U/mL). Network analysis revealed Lactobacillaceae are closely linked with saccharifying communities. This study establishes microenvironmental gradients as critical regulators of spatial saccharification in *Daqu*, informing strategies to optimize microbial consortia for *baijiu* production.

## 1. Introduction

Solid-state fermentation (SSF) ecosystems drive the production of foods like soy sauce, cheese, and *baijiu* (Chinese liquor) [1]. Within these systems, microbial metabolism critically defines product quality, yield, and safety [2,3]. Despite ongoing technological advances, ubiquitous challenges persist in these fermented products: environmental gradients of temperature, moisture, oxygen, and other abiotic factors arising within the product structure result in significant spatial stratification of resident microbial communities [4,5,6], compromising product uniformity and obscuring links between genetic potential (e.g., key functional genes) and metabolic outputs like enzyme activity [7,8].

Strong-flavor *baijiu*, a traditional Chinese distilled liquor of profound cultural and economic significance, relies on *Daqu*, an SSF starter supplying saccharifying enzymes and core microbiota [9,10]. *Daqu* governs starch saccharification and the flavor precursor formation [11,12], with saccharifying enzymes underpinning ethanol synthesis and flavor development [13,14].

Although microbial community succession in *Daqu* fermentation is well studied [15,16], a key gap in how abiotic gradients across *Daqu* layers (QP, HQ, QX) regulate spatial distributions of critical functional microorganisms (e.g., saccharifying consortia) remains [17,18]. Similar gradients are exerted in cheese or soy sauce moromi [19,20]. Such gradients undoubtedly exert strong selective pressures on microbes, likely shaping their biogeographical patterns [21], yet their collective impact on enzyme spatiotemporal dynamics, microbial interactions, and functional stratification in *Daqu* is uncharacterized. Past studies often treated *Daqu* as homogeneous [22], overlooking internal niche stratification that may drive functional differentiation [23].

Building on this background, the study aims to elucidate how microenvironmental gradients in strong-flavor *Daqu* drive spatial stratification of saccharifying enzymes and microbiota across distinct layers (QP, HQ, QX), ultimately compartmentalizing metabolic functions. To address this, this study integrated physicochemical profiling, metagenomics, and PacBio full-length sequencing to identify key saccharifying enzymes and their microbial sources, to quantify abiotic microbial regulation of enzyme gene abundance and activity, and to resolve the spatial architecture of saccharification functional guilds and their ecological interactions across strata.

This mechanistic understanding will establish a robust scientific foundation for optimizing baijiu production through precision management of microbial consortia based on spatial environmental parameters.

## 2. Materials and Methods

### 2.1. Sample Collection

Strong-flavor *Daqu* samples were collected from a production workshop in Sichuan Province, China, in November 2023. Sampling occurred at ten time points across two distinct phases: the fermentation phase (Days 0, 2, 4, 8) and the storage phase (Days 12, 20, 30, 50, 70, 90). At each time point, nine independent *Daqu* bricks were randomly selected. Each brick was sectioned into three layers defined as follows: QP-surface layer (~1 cm thickness), HQ-subsurface layer (1–2.5 cm depth, distinguished by dark brown pigmentation in matured *Daqu*), and QX-core layer (remaining central section). The depth boundaries (QP: 0–1 cm; HQ: 1–2.5 cm; QX: >2.5 cm) were established based on the gradual appearance of characteristic dark brown bands within the 1–2.5 cm depth range of the *Daqu* during fermentation. Figure 1 shows the three layers of strong-flavor *Daqu*, illustrating the characteristic dark brown pigmentation defining the HQ layer. The 0-day samples were retrieved from the conveyor belt. QP, HQ, and QX samples (due to the high moisture content in the 0 d samples, they cannot be crushed; therefore, they were mixed) were crushed. All samples were split into two portions: one portion stored at −20 °C for physicochemical analysis, the other portion stored at −80 °C for further assays.

### 2.2. Analysis of Environmental and Physicochemical Properties

The saccharifying activity, temperature, moisture content, and titratable acidity of strong-flavor *Daqu* were measured following the methods described by QB/T 4257−2011 (General Analytical Methods for Daqu of Chinese Spirits) [24]. All properties were determined in triplicate and calculated on a dry-weight basis.

### 2.3. Starch Hydrolysis Efficiency by Distinct Saccharifying Enzymes

Starch Solution: A 2% (*w*/*v*) starch solution was prepared using wheat flour (*Daqu* raw material), sorghum flour (*baijiu* raw material), and soluble starch (Chinese National Standards substrate; QB/T 4257−2011).

Enzyme Solutions: Glucan 1,4-α-glucosidase (EC 3.2.1.3), α-amylase (EC 3.2.1.1), and α-glucosidase (EC 3.2.1.20) were diluted to 50 U/g (to guarantee substrate saturation throughout enzymatic reactions). Enzymes EC 3.2.1.3, EC 3.2.1.1, and EC 3.2.1.20 were purchased from Henan Zhongcheng Biotechnology Co., Ltd. (Henan, China), Tianjin Zhonglian Chemical Reagent Co., Ltd. (Tianjin, China), and Zhejiang Mingzhe Biotechnology Co., Ltd. (Zhejiang, China), respectively.

Enzymatic Assay: Experimental Groups: Enzyme combinations followed Table 1 (5 groups). Control reactions contained substrate without enzymes to assess non-enzymatic hydrolysis and heat-inactivated enzymes (100 °C, 10 min) to confirm activity loss. We incubated 10 mL starch solution + enzymes at 35 °C for 60 min. Saccharifying activity was determined following QB/T 4257−2011.

### 2.4. Verification of Saccharifying Activity in Enzyme-Producing Functional Strains

Strain origin: The main enzyme-producing strains are sourced from the Luzhou Laojiao Culture Collection, including *Lichtheimia ramose*, *Aspergillus oryzae*, *Paecilomyces variotii*, and *Rasamsonia composticola*.

Seed culture preparation: Strains were inoculated into potato dextrose broth (PDA) and cultured at 27 °C with 150 rpm orbital shaking for 48 h to obtain seed cultures.

Solid-state fermentation: Seed culture (1 mL) was inoculated into wheat-based solid-state fermentation media (10 g wheat flour homogenized with 8 mL distilled water). Cultures were incubated at 27 °C for 2.5 days. Saccharifying activity was measured against heat-inactivated controls (autoclaved seed cultures).

Enzymatic analysis: saccharifying activity was quantified using the 3,5-dinitrosalicylic acid (DNS) method for reducing sugar determination [25].

### 2.5. DNA Extraction

The extraction of microbial genomic DNA from strong-flavor *Daqu* was performed by Shanghai Meiji Biological Co., Ltd. (Shanghai, China). The integrity of the DNA was assessed using 1% agarose gel electrophoresis; the purity and concentration of the DNA were evaluated with a NanoDrop2000 and a Quantus Fluorometer (Picogreen) (Shanghai, China).

### 2.6. PacBio Full-Length Diversity Sequencing

The bacterial 16S rRNA V1-V9 region was amplified using primers 27F/1492R, and the fungal ITS region using primers ITS1F/ITS4R. Reactions (20 µL; Pro Taq Master Mix) contained each primer (0.8 µL, 5 µM), template DNA (10 ng), and nuclease-free water. Cycling conditions (ABI GeneAmp^®^ 9700, Thermo Fisher Scientific Co., Ltd, Shanghai, China): 95 °C (3 min); 30 cycles of 95 °C (30 s), 55 °C (30 s), 72 °C (45 s); 72 °C (10 min). Triplicate reactions per sample were pooled, verified by 2% agarose gel, and purified (AxyPrep DNA Gel Extraction Kit, Axygen, Hangzhou, China). Purified amplicons were quantified (QuantiFluor™-ST, Promega, Beijing, China) and pooled equimolarly. PacBio SMRTbell libraries were constructed (end repair, blunt adapter ligation, exonuclease purification). Sequencing was performed on the PacBio Sequel IIe platform (Shanghai Meiji, Shanghai, China) using HiFi Circular Consensus Sequencing (CCS) chemistry. Raw HiFi reads (CCS ≥ Q20) were generated using SMRT Link v11.0, demultiplexed, quality-filtered, and oriented. Operational Taxonomic Units (OTUs) were clustered at 97% nucleotide identity (Uparse v7.0.1090), filtering chimeras and singletons. Taxonomic assignment of OTU representatives employed the RDP classifier (v2.13, Bayesian algorithm) against the NT database. Inadequate 16S rRNA PCR amplicons from HQ30D *Daqu* samples excluded sequencing, likely due to DNA degradation during transport, storage, or extraction.

### 2.7. Metagenomic Sequencing

Genomic DNA was extracted from *Daqu* samples and randomly fragmented to 300 bp (Covaris M220). Indexed libraries were constructed per sample. Libraries were pooled and sequenced on the Illumina NovaSeq platform (Shanghai Meiji Biological Co., Ltd. Shanghai, China) using paired-end 150 bp (PE150) chemistry. Raw reads were demultiplexed based on sample-specific indexes. Quality control, including adapter removal and quality trimming/filtering, was performed using Fastp (v0.20.1), resulting in high-quality reads (Appendix A). Residual host DNA contamination was removed by aligning reads (BWA, v0.7.17) against the host reference sequence. Filtered reads from each sample were assembled de novo using MEGAHIT (v1.1.2), retaining contigs ≥ 300 bp. Open reading frames (ORFs) were predicted from contigs using Prodigal (v2.6.3), selecting ORFs encoding proteins ≥ 100 amino acids. Predicted genes across all samples were clustered at 95% nucleotide identity and 90% coverage (CD-HIT, v4.6.1). The longest sequence within each cluster was selected to form a non-redundant gene catalog. Quality-controlled reads per sample were mapped back to the non-redundant gene catalog (SOAPaligner v2.21). Gene abundance (RPKM: Reads Per Kilobase fragment length per Million mapped reads) per sample was calculated. Non-redundant genes were annotated against the NCBI NR protein database (DIAMOND, v2.0.13; blastp, e-value ≤ 1 × 10^−5^). Taxonomic lineage was derived from NR-associated information. Species abundance was calculated as the sum of RPKM values of its annotated genes. Genes were annotated against the KEGG database (DIAMOND, blastp, e-value ≤ 1 × 10^−5^). Pathway/Enzyme abundance was calculated as the total RPKM of its annotated genes. Genes were annotated against the CAZy database v5.0 (hmmscan, e-value ≤ 1 × 10^−5^). CAZyme family abundance was calculated as the total RPKM of its annotated genes.

### 2.8. Statistical Analysis

The line chart, bubble chart, stacked histograms, and heatmap were all drawn using OriginPro 2024b. Variations in physicochemical properties were assessed via one-way ANOVA followed by Tukey’s HSD post hoc test for pairwise comparisons (SPSS 26.0, *p* < 0.05). Spearman correlations between genera of QP, HQ, and QX species (|r| > 0.7, *p* < 0.05) were computed, and correlation network diagrams were constructed using the online tool at https://www.omicstudio.cn/tool (accessed on 10 September 2025). The graphs were combined using Adobe Illustrator (version 2021).

## 3. Results

### 3.1. Analysis of Physicochemical Properties of Strong-Flavor Daqu

Saccharifying activity declined sharply across all *Daqu* layers after an initial peak (days 0–2, Figure 2a). Activity later recovered during storage (day 12 onward), stabilizing by day 30. This phase (12–30 d) defined a critical period for impacting final saccharifying activity. Saccharifying activity decreased from QP (highest) to QX (lowest) in a gradient, and this spatial divergence corresponded to microenvironmental gradients (*p* < 0.05). Moisture content decreased progressively across layers (QX > HQ > QP) (Figure 2b). Temperature initially peaked then declined, with QX consistently higher than QP and HQ (Figure 2c). Titratable acidity in QX exceeded that of QP and HQ during days 0–20, then declined (Figure 2d). Since saccharifying enzymes originate primarily from microbial metabolism [26], spatial variations in physicochemical properties may drive differential colonization by functional microbiota [27,28,29].

### 3.2. Saccharification Pathway and Key Saccharifying Enzymes of Strong-Flavor Daqu

The saccharifying activity of *Daqu* is its ability to hydrolyze starch into reducing sugars [30]. Metagenomic sequencing was conducted on samples of the QP, HQ, and QX during the fermentation process to analyze the starch hydrolysis pathway, and referring to the Pathway map of Starch and sucrose metabolism in the KEGG database, to map the starch hydrolysis pathway (Figure 3a). Five principal enzymes were involved in the starch hydrolysis process (Figure 3b), the average gene abundance ranked from highest to lowest is α-glucosidase (EC 3.2.1.20), α-amylase (EC 3.2.1.1), oligo-α-glucosidase (EC 3.2.1.10), glucan 1,4-α-glucosidase (EC 3.2.1.3), and maltose phosphorylase (EC 2.4.1.8). Enzymes (glucan 1,4-α-maltohydrolase (EC 3.2.1.133), β-amylase (EC 3.2.1.2), and isoamylase (EC 3.2.1.68)) with negligible abundance (<0.1% of total saccharifying enzyme RPKM) were excluded from further analysis.

Days 12–30 represent the most significant phase for increased saccharifying activity in *Daqu* (Figure 2a). However, during this period, gene abundances of EC 3.2.1.10 and EC 2.4.1.8 remained consistently low in both the HQ and QX. This implies that these enzymes likely contribute minimally to saccharification efficiency and are not key enzymes in starch hydrolysis. EC 3.2.1.3, EC 3.2.1.1, and EC 3.2.1.20 may be potential key saccharifying enzymes in strong-flavor *Daqu*, though further investigation is required.

In vitro validation suggests that combining EC 3.2.1.1 and EC 3.2.1.20 could elevate saccharifying activity significantly (*p* < 0.05; Figure 3c). In contrast, adding EC 3.2.1.3 did not enhance activity significantly (*p* > 0.05), suggesting EC 3.2.1.1 and EC 3.2.1.20 are primary drivers.

### 3.3. Microbial Sources of Key Saccharifying Enzymes in Strong-Flavor Daqu

Based on the KEGG database, annotation analysis was performed to identify microbial sources of key saccharifying enzymes in strong-flavor *Daqu*.

Distinct spatiotemporal dynamics characterized microbial contributors to key saccharifying enzymes across *Daqu* (Figure 4). For EC 3.2.1.1, bacterial genera *Bacillus* and *Lactiplantibacillus* dominated initial fermentation (0–4 d), shifting to fungal dominance post day 8. *Lichtheimia* and *Aspergillus* prevailed in QP and HQ layers, whereas QX exhibited mid-stage contributions from *unclassified Fungi* and *Paecilomyces*, shifting to *Rasamsonia* and *Kroppenstedtia* later (Figure 4a).

The EC 3.2.1.20 gene was mainly annotated to *Lichtheimia* in QP, and the abundance of *Aspergillus* increased gradually with fermentation time (Figure 4b). In HQ and QX, the EC 3.2.1.20 gene was mainly annotated to *Lactiplantibacillus*, *Companilactobacillus*, and *Paucilactobacillus* from days 2 to 4. Subsequently, the HQ was mainly annotated to *Lichtheimia* and *Fennellomyces*, and the QX was mainly annotated to *Aspergillus*, *Paecilomyces*, and *Rasamsonia*. However, the abundance of *Paecilomyces* and *Rasamsonia* gradually decreased and approached zero in QX during the later stages of fermentation.

### 3.4. Driving Factors of Microbial Saccharifying Enzymes Gene Abundance Changes in Strong-Flavor Daqu

Functional gene abundance, indicative of microbial functional potential yet environmentally labile [31,32], was analyzed via correlation network to elucidate relationships between environmental factors and saccharifying enzyme gene abundances (EC 3.2.1.1, EC 3.2.1.20) across *Daqu* layers (QP, HQ, QX), revealing microbial environmental feedback (Figure 5).

Spearman correlation analysis revealed that titratable acidity was positively correlated with both enzyme genes in QP and QX, specifically associating *Lactiplantibacillus* (QP) and *Lederbergia* (QX) with EC 3.2.1.1 and *Lacticaseibacillus* (QP) and *Levilactobacillus* (QX) with EC 3.2.1.20. While temperature overall negatively correlated with target gene abundances, a distinct positive correlation emerged between QX temperature and EC 3.2.1.20 expression within thermotolerant genera (*Lactiplantibacillus*, *Companilactobacillus*, and *Weissella*). Moisture content generally suppressed enzyme genes across layers, though EC 3.2.1.1 uniquely increased with moisture in QP. Critically, environmental sensitivity escalated progressively from QP to QX, with QX temperature impacting 24 microbial taxa and moisture affecting 34 taxa, demonstrating intensifying environmental constraints towards the core region.

The saccharifying gene abundance (RPKM) indicates metabolic potential, but environmental screening (e.g., QX’s high heat or humidity) may disrupt translation or post-translational modification, decoupling genomics from functional output [33,34]. To verify whether environmental factors influence the saccharifying activity of microorganisms, the saccharifying activity of major enzyme-producing microorganisms was determined (Figure 5g). Strains predominantly localized in QP and HQ layers (*Lichtheimia ramose*, *Aspergillus oryzae*) demonstrated significantly higher saccharifying activity. In contrast, QX-associated strains (*Paecilomyces variotii*, *Rasamsonia composticola*) showed markedly reduced activity levels. This suggests microenvironmental extremes in QX may suppress enzymatic efficiency or protein stability, even when genetic potential exists.

### 3.5. Succession of Saccharifying Enzyme-Producing Microorganisms in Strong-Flavor Daqu

The microbial sources of two amylases, EC 3.2.1.1 and EC 3.2.1.20 (Figure 4), were identified, and their corresponding taxonomic abundances (at the genus level) were determined from PacBio full-length diversity sequencing data. A heatmap was generated (Figure 6), where the species were arranged from bottom to top in descending order of relative abundance.

Analysis of the bacterial community structure revealed that QP-associated microorganisms maintained dynamic stability throughout fermentation, while HQ-enriched taxa peaked transiently during early maturation (2–8 d). In QX, EC 3.2.1.20-producing bacteria were ephemeral early colonizers (2–8 d), whereas EC 3.2.1.1 producers persisted with fluctuating abundance. For the fungal community, QP demonstrated rapid colonization (0–12 d) by stable dominants *Aspergillus* and *Saccharomycopsis*, HQ accrued fungal biomass primarily in mid-late stages (8–90 d), and QX showed bifurcated activity peaks during initial (0–2 d) and terminal maturation phases (30–90 d). Bacterially dominated consortia (*Companilactobacillus*, *Lactiplantibacillus*, and *Weissella*) characterized QP during the critical saccharification elevation phase (12–30 d), whereas *Bacillus*, *Aspergillus*, and *Saccharomycopsis* assemblages prevailed simultaneously in HQ and QX.

### 3.6. Co-Occurrence Network Analysis of Microorganisms in Strong-Flavor Daqu

In this study, the top 15 bacterial and 15 fungal genera (totaling 30 genera) were selected to construct microbial interaction networks. Spearman correlation analysis was employed to evaluate the strength of microbial associations, with filtering criteria set at an absolute correlation coefficient |r| > 0.7 and significance *p* < 0.05. Microbial correlation analyses were performed for the fermentation phase (0–8 days) and the storage phase (12–90 days) (Figure 7).

The fermentation phase exhibited pervasive positive correlations among bacteria across all layers; except for HQ, the storage phase networks displayed increased complexity with significantly augmented negative correlations (*p* < 0.05, |r| > 0.7). QP transitioned from cooperative bacterial clusters (*Amnimonas*, *Brevundimonas*, and *Pseudomonas*) to intensified competition among high enzyme producers (*Aspergillus*, *Furfurilactobacillus*, and *Lactiplantibacillus*); HQ shifted from *Brevundimonas*-dominated bacteria to storage phase fungal–bacterial hubs (*Furfurilactobacillus* and *Kluyveromyces*); and QX evolved from early synergistic alliances (*Penicillium*, *Streptococcus*, and *Mammaliicoccus*) to storage phase associations dominated by *Phyllobacterium*, *Brevundimonas*, *Rasamsonia*, and *Pichia*.

In the fermentation stage, the interactive network among saccharifying functional microorganisms in the QP and HQ areas was sparse, while QX exhibited a strongly synergistic network centered on *Penicillium*, *Streptococcus*, and *Mammaliicoccus*. Notably, Lactobacillaceae showed a significant positive correlation with the saccharifying microbial community during the storage phase.

## 4. Discussion

Saccharification critically governs *baijiu* yield and quality [30], yet the spatiotemporal dynamics of key enzymes within stratified *Daqu* remain poorly characterized.

A pronounced gradient was observed in saccharifying activity among *Daqu* layers (QP > HQ > QX, Figure 2a). This decline correlates inversely with escalating temperature and moisture toward the center (Figure 2b,c) and aligns with differential abiotic stress across layers. Sustained high temperatures (Figure 2c) in QX are likely responsible for the observed lower enzymatic activity compared to QP strains, despite comparable genomic potential (Figure 4). Mechanistically, these elevated temperatures can denature key saccharifying enzymes [35]. Such abiotic filtering shapes microbial biogeography: QP and HQ host high-activity fungi (e.g., *Lichtheimia*, *Aspergillus*), whereas thermotolerant but low-activity taxa (e.g., *Paecilomyces*, *Rasamsonia*) dominate QX (Figure 4 and Figure 5g), consistent with prior findings on abiotic filtering in SSF ecosystems [23,29,36].

Metagenomics identified EC 3.2.1.1 and EC 3.2.1.20 as the key saccharifying enzymes (Figure 3), and *Lichtheimia* (QP and HQ) emerges as their dominant microbial source (Figure 4), consistent with its superior in vitro activity (Figure 5g). These findings contrast with reports of prior homogenized *Daqu* studies highlighting EC 3.2.1.3, EC 3.2.1.1, and EC 3.2.1.20 [37,38]. This discrepancy is possibly due to variations in omics technology and types of *Daqu* [22,30]. While EC 3.2.1.3 enhanced saccharification synergistically, its kinetic limitation against α-1,6 glycosidic bonds restricted standalone efficacy [39] (Figure 3c and Figure A1). Notably, EC 3.2.1.20’s role beyond maltose hydrolysis warrants further investigation, given its synergistic potential with EC 3.2.1.1 [20,40].

Saccharifying microbial communities displayed distinct spatiotemporal succession patterns (Figure 4). In QP and HQ, bacterial dominance (e.g., *Bacillus*, *Lactiplantibacillus*) during early fermentation (0–4 d) shifted to fungal stability (*Lichtheimia*, *Aspergillus*) in maturation phases, facilitated by higher oxygen flux and lower thermal inertia [35]. In QX, early bacterial ephemerality (e.g., *Companilactobacillus*) gave way to thermotolerant genera (*Rasamsonia*, *Kroppenstedtia*) under sustained heat stress. Variances in the environment across layers of *Daqu* could result in disparities in the abundance of saccharifying enzyme genes (Figure 5) [36]. This illustrates microenvironment-driven functional divergence—consistent with studies comparing *Daqu* types across fermentation regimes [13].

In vitro validation revealed significantly lower saccharifying activity in QX-associated strains (e.g., *Paecilomyces variotii*; Figure 5g) compared to QP and HQ strains (e.g., *Lichtheimia ramosa*), despite higher metagenomic gene abundance of *Paecilomyces* in QX (Figure 4a,b). This apparent paradox highlights a decoupling between genetic potential and functional output in the QX layer, which experiences extreme microenvironments (high heat and humidity; Figure 2b,c). While the precise mechanism remains unresolved, environmental stressors in QX likely disrupt enzyme functionality through thermal denaturation, pH instability, or substrate limitations.

Co-occurrence Network analyses revealed cooperative interactions in the fermentation phase (positive correlations), transitioning to competitive exclusion storage phase (negative correlations). This ecological shift aligns with resource competition models, wherein depleted substrates intensify negative interactions among saccharifying genera [41,42]. But the HQ compartment experiences reduced competition in the later stages of fermentation (Figure 7c,d); the reason may be that the hydroscopic starch hydrolysis in this compartment is the fastest, promoting the build-up of reducing sugars (Figure A2), thus increasing the available carbon source for microbial growth. This resource abundance reduces competitive pressure among microorganisms, consequently weakening ecological interactions and resulting in diminished microbial correlation coefficients during later fermentation stages. Notably, Lactobacillaceae are closely linked with saccharifying communities, extending beyond mere acidification. They may symbiotically interact with saccharifying microorganisms by consuming oxygen, producing acids to inhibit competitors, and establishing a low-pH anaerobic environment, thereby indirectly enhancing the stability of saccharifying communities and promoting synergistic fermentation processes [43,44].

While spatial stratification is integral to *Daqu* function, certain limitations exist in this study. These spatial patterns were observed in a November fermentation batch, future multi-seasonal sampling is needed to confirm their robustness across operational variations. Seasonal shifts in microbial guilds necessitate multiseason validation to assess temporal stability [45]. Nevertheless, the spatial patterns align with known physicochemical heterogeneities in *Daqu* [17,18], suggesting that microenvironmental compartmentalization is an inherent feature of fermentation. While gene abundances in QX (e.g., *Paecilomyces*) suggested high saccharification potential, the functional decoupling noted the need for metaproteomics to quantify actual enzyme expression [13,30]. While strain-level activity assays confirmed functional differences, enzymatic kinetics (e.g., Km/Vmax) were not characterized. Future work should address kinetic parameters to fully resolve catalytic efficiency.

The results provide strategies for precision modulation of *Daqu* fermentation. Augmenting QP and HQ zones with high-yield strains (e.g., *Lichtheimia ramosa*) may boost saccharification. Regulating moisture dispersion and thermal buffering in center layers could mitigate QX functionality loss. Such strategies align with emerging efforts to engineer synthetic microbial consortia for standardized *Daqu* production.

## 5. Conclusions

This study establishes that saccharifying activity in strong-flavor *Daqu* exhibits a significant spatial gradient (QP > HQ > QX), driven by microenvironmental heterogeneity. Metagenomic and enzymatic analyses identified α-amylase and α-glucosidase as key saccharifying enzymes, predominantly sourced from the fungus *Lichtheimia* in QP and HQ layers. Environmental factors critically modulated enzyme gene abundance: temperature and moisture suppressed its abundance (notably in QX), while titratable acidity enhanced it. Strains from QP and HQ demonstrated superior activity over QX adapted taxa. Lactobacillaceae are closely linked with saccharifying communities. This layered functionality informs interventions for *Daqu* optimization. Future work will focus on seasonality validation and metaproteomic verification of enzyme expression dynamics.

## Figures and Tables

**Figure 1 foods-14-04160-f001:**
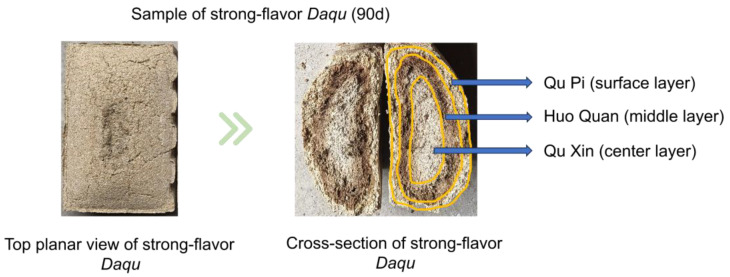
The three layers of strong-flavor *Daqu*.

**Figure 2 foods-14-04160-f002:**
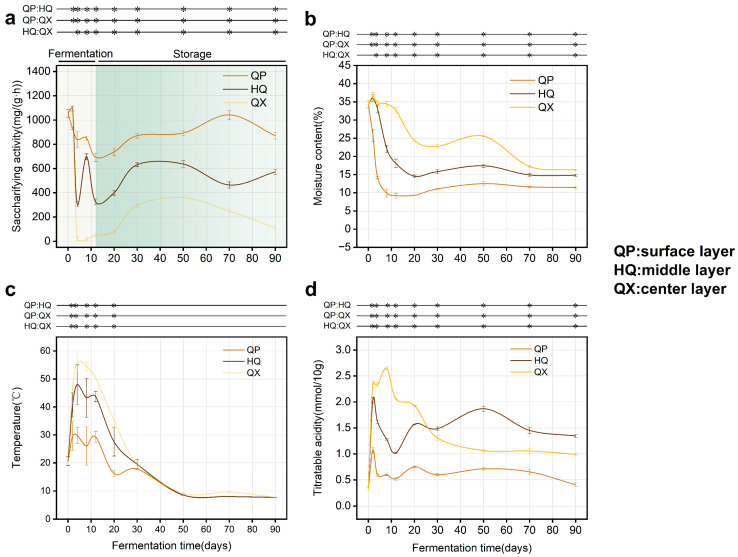
Dynamic changes of physicochemical properties during fermentation (*n* = 3), * denotes significant differences (*p* < 0.05, one-way ANOVA, Tukey’s HSD test). (**a**) saccharifying activity; (**b**) moisture content; (**c**) temperature; (**d**) titratable acidity.

**Figure 3 foods-14-04160-f003:**
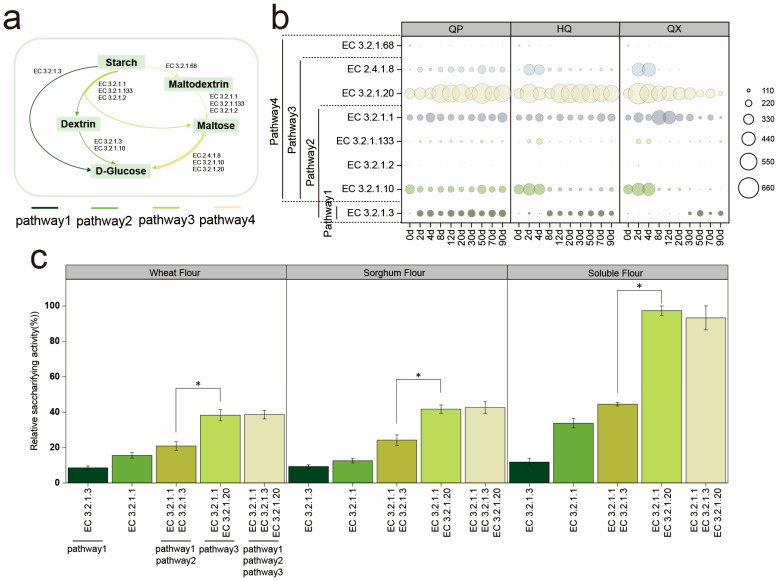
(**a**) Starch hydrolysis pathway. (**b**) The changes of saccharifying enzymes gene abundances (RPKM value) in strong-flavor *Daqu*. (**c**) The saccharifying activity of different saccharifying enzymes. * denotes significant differences (*p* < 0.05, one-way ANOVA, Tukey’s HSD test).

**Figure 4 foods-14-04160-f004:**
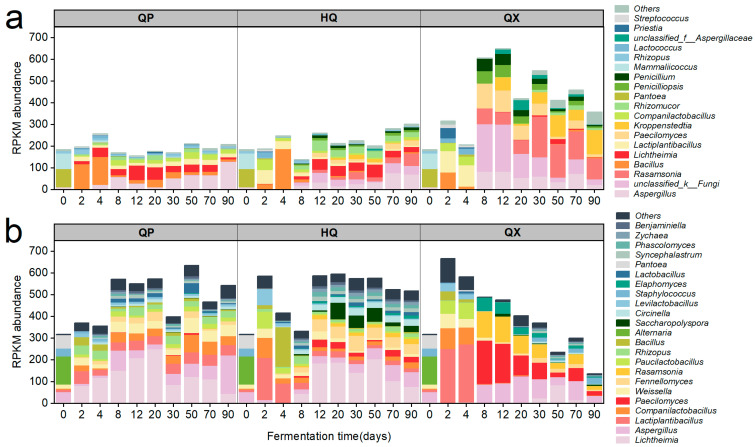
Microbial sources of saccharifying enzymes (genus level), (**a**) EC 3.2.1.1 and (**b**) EC 3.2.1.20.

**Figure 5 foods-14-04160-f005:**
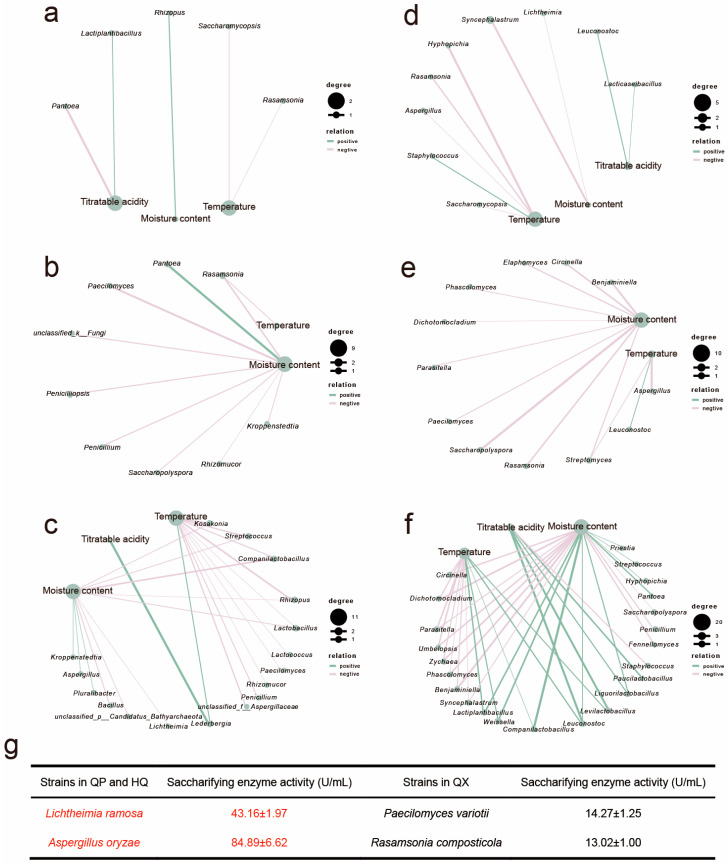
The correlation network analysis between EC 3.2.1.1 and EC 3.2.1.20 gene abundances and environmental factors in different microorganisms. (r > 0.7, *p* < 0.05), EC 3.2.1.1: (**a**) QP, (**b**) HQ, (**c**) QX; EC 3.2.1.20: (**d**) QP, (**e**) HQ, (**f**) QX. (**g**) Saccharifying enzyme activity of enzyme-producing functional microorganisms.

**Figure 6 foods-14-04160-f006:**
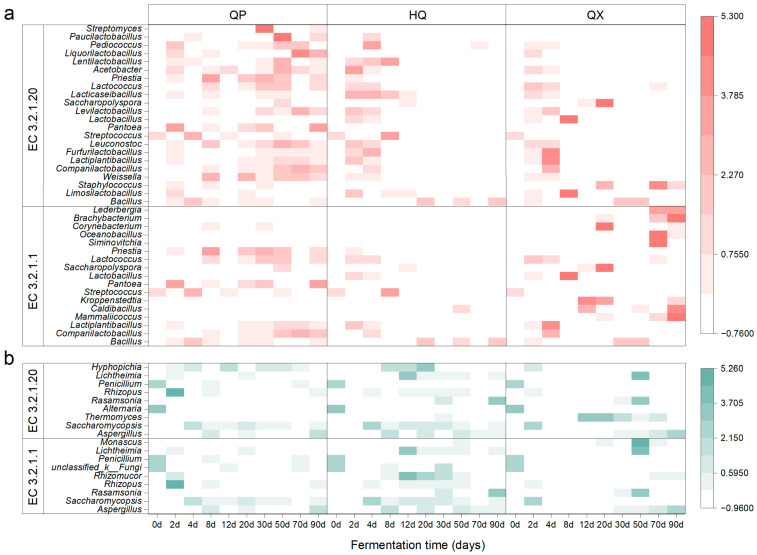
Succession of saccharifying enzyme-producing microorganisms in strong-flavor *Daqu* (genus level), (**a**) bacteria and (**b**) fungi.

**Figure 7 foods-14-04160-f007:**
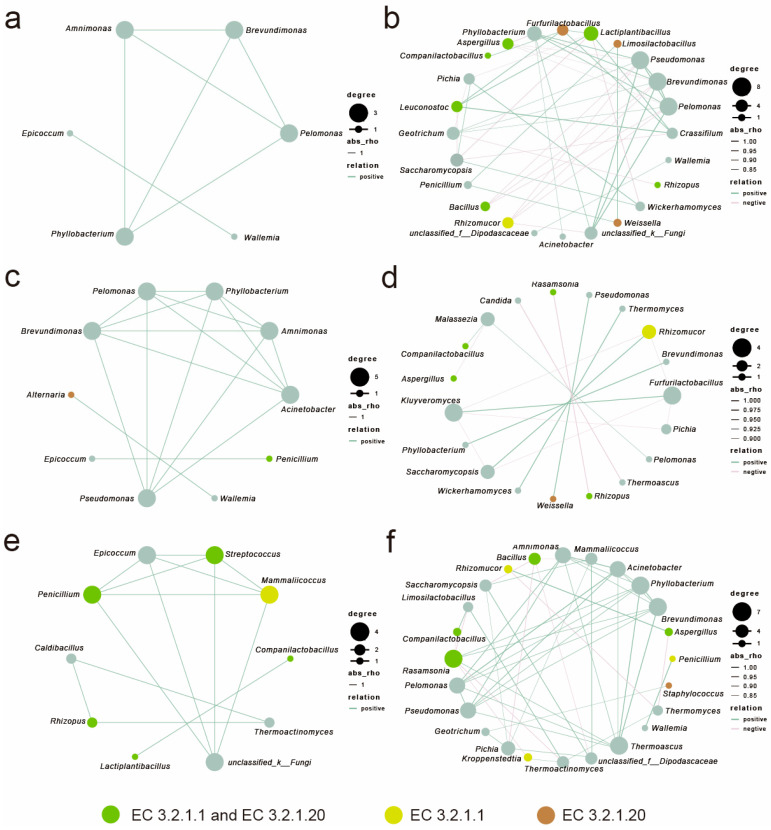
Co-occurrence network analysis of microorganisms in strong-flavor *Daqu* at the genus level (r > 0.7, *p* < 0.05). QP: fermentation phase (**a**), storage phase (**b**); HQ: fermentation phase (**c**), storage phase (**d**); QX: fermentation phase (**e**), storage phase (**f**).

**Table 1 foods-14-04160-t001:** The amount of enzyme added in each group.

Enzyme	Group				
	Group 1	Group 2	Group 3	Group 4	Group 5
EC 3.2.1.3	1 mL		1 mL		1 mL
EC 3.2.1.1		1 mL	1 mL	1 mL	1 mL
EC 3.2.1.20				1 mL	1 mL

## Data Availability

The original contributions presented in this study are included in the article/Appendix A; further inquiries can be directed to the corresponding authors.

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
