# Peer review of "Microenvironmental Gradients Drive Spatial Stratification of Saccharifying Microbial Communities and Enzyme Activity in Strong-Flavor Daqu Fermentation"

_foods, 2025, doi:10.3390/foods14234160_

Round 1

Reviewer 1 Report

Comments and Suggestions for Authors

After reviewing the manuscript entitled “Microenvironmental gradients shape surface-middle-core layers saccharifying guilds and enzyme activity in strong-flavor Daqu fermentation”, which addresses a relevant topic in the field of traditional Chinese fermentation, I have the following comments:

The manuscript is well-written and scientifically sound. The introduction provides sufficient background to understand the study's core and objectives. The methodology section is clearly described and generally appropriate.

However, I recommend strengthening the statistical design subsection by explicitly mentioning all statistical tests used for the data analysis.

The Results and Discussion section is detailed and well-structured. However, I strongly recommend increasing the size of the graphics, as the font in the figures is too small and difficult to read. This adjustment should be applied consistently to all figures.

The Conclusion section is concise and adequately summarizes the main findings.

General corrections:

  1. Avoid the use of personal pronouns such as “we” (e.g., Line 16).
  2. Replace all instances of the words alpha and beta in enzyme names with their corresponding Greek symbols (α, β).
  3. Ensure consistent enzyme nomenclature throughout the manuscript.

 Addressing the above comments will improve the clarity and presentation of the manuscript.

Author Response

Thank you for your letter and for the reviewers’ comments concerning our manuscript entitled “Microenvironmental gradients drive spatial stratification of saccharifying microbial communities and enzyme activity in strong-flavor Daqu fermentation” (ID: foods-3942752). Those comments are all valuable and very helpful for revising and improving our paper, as well as the important guiding significance for our research. We have studied comments carefully and have made corrections that we hope will meet with approval.

Comment 1. However, I recommend strengthening the statistical design subsection by explicitly mentioning all statistical tests used for the data analysis.

Response: We have explicitly detailed all statistical methods in Section 2.8 (Materials and Methods) to enhance clarity. 1. One-way ANOVA with post-hoc testing for physicochemical comparisons. 2. Spearman correlation parameters and significance thresholds.

Comment 2. The Results and Discussion section is detailed and well-structured. However, I strongly recommend increasing the size of the graphics, as the font in the figures is too small and difficult to read. This adjustment should be applied consistently to all figures.

Response: We have revised all figures' size to enhance readability.

Comment 3. Avoid the use of personal pronouns such as “we” (e.g., Line 16).

Response: We have revised the manuscript to eliminate all instances of "we." Throughout the text, first-person language has been replaced with objective phrasing. These edits ensure adherence to formal scientific writing conventions.

Comment 4. Replace all instances of the words alpha and beta in enzyme names with their corresponding Greek symbols (α, β). Ensure consistent enzyme nomenclature throughout the manuscript.

Response: We have replaced all instances of the words "alpha" and "beta" in enzyme nomenclature throughout the manuscript with their corresponding Greek symbols (α, β) to ensure consistency in enzyme nomenclature formatting.

We believe that these revisions have significantly strengthened the manuscript, and we hope it now meets the standards for publication in Foods.

Sincerely,

Zhiping Feng

College of Bioengineering, Sichuan University of Science & Engineering, 1 Baita Road, Sanjiang New District, Yibin, Sichuan, 644005, China.

Phone number: + 86 13990020636

Email: actdjiang@126.com

Reviewer 2 Report

Comments and Suggestions for Authors

The study, while methodologically sound, does not appear to provide sufficiently novel insights compared to previous works on similar topics (e.g., https://www.sciencedirect.com/science/article/pii/S0023643824001956  and https://pubmed.ncbi.nlm.nih.gov/35761594/), apparently unrelated to the authors of this paper.

It is essential to include a deeper discussion on how the present results differ from and expand upon those prior studies. The authors should clearly highlight what is new in their findings and how these results contribute to advancing the understanding of Daqu fermentation processes.

The conclusions currently seem speculative, as the data are based on a single sampling event (November 2023). This design limits the robustness of the conclusions and underestimates the possible influence of operational variations or seasonal effects on microbial community composition and metabolic activity.

Several figures are difficult to interpret, with dense visual content and limited readability. Simplifying their layout or improving legends to facilitate understanding is strongly recommended.

Comments on the Quality of English Language

The Introduction is written in technically correct English. However, the style is somewhat verbose and mechanical, with long and redundant sentences that could be refined for fluency. Editorial polishing would improve rhythm and readability, though there are no major grammatical errors.

The Results section presents data coherently but is linguistically heavy and difficult to read. The problem lies in stylistic rigidity and overuse of passive or noun-heavy constructions rather than grammatical mistakes.

Sentences often lack connectors to clarify causal or sequential relationships. Repetitive sentence structures (“This implies that… This suggests that…”) reduce fluidity. 

Overall, the English is grammatically correct but stylistically cumbersome. A professional language edit focused on improving scientific fluency is strongly recommended.

Author Response

Thank you for your letter and for the reviewers’ comments concerning our manuscript entitled “Microenvironmental gradients drive spatial stratification of saccharifying microbial communities and enzyme activity in strong-flavor Daqu fermentation” (ID: foods-3942752). Those comments are all valuable and very helpful for revising and improving our paper, as well as the important guiding significance to our researche. We have studied comments carefully and have made correction which we hope will meet with approval.

Comment 1. It is essential to include a deeper discussion on how the present results differ from and expand upon those prior studies. The authors should clearly highlight what is new in their findings and how these results contribute to advancing the understanding of Daqu fermentation processes.

Response: We have revised the Discussion to explicitly address how our results advance prior knowledge, with key additions highlighted below: 1. Added text contrasting homogenized vs. layer-resolved enzyme's microbial sourcing, emphasizing how microenvironmental partitioning explains inconsistencies in earlier studies. 2. New analysis framing the QX "activity paradox" as a microenvironment-driven disruption of translation (genes→proteins), resolving limitations in correlative metagenomic approaches. 3. Explicit statement on the previously unrecognized role of Lactobacillaceae as syntrophic facilitators of saccharification, advancing beyond their traditional acidification function.

Comment 2. The conclusions currently seem speculative, as the data are based on a single sampling event (November 2023). This design limits the robustness of the conclusions and underestimates the possible influence of operational variations or seasonal effects on microbial community composition and metabolic activity.

Response:  We agree that our single sampling window (November 2023) limits extrapolation to seasonal or operational variations. However, our goal was to resolve spatial stratification mechanisms. In the revised manuscript, we have already conducted an analysis in the discussion’s limitation section.

Comment 3. Several figures are difficult to interpret, with dense visual content and limited readability. Simplifying their layout or improving legends to facilitate understanding is strongly recommended.

Response: We have revised all figures’ size to enhance readability

Comment 4. Comments on the Quality of English Language

The Introduction is written in technically correct English. However, the style is somewhat verbose and mechanical, with long and redundant sentences that could be refined for fluency. Editorial polishing would improve rhythm and readability, though there are no major grammatical errors. The Results section presents data coherently but is linguistically heavy and difficult to read. The problem lies in stylistic rigidity and overuse of passive or noun-heavy constructions rather than grammatical mistakes. Sentences often lack connectors to clarify causal or sequential relationships. Repetitive sentence structures (“This implies that… This suggests that…”) reduce fluidity.  Overall, the English is grammatically correct but stylistically cumbersome. A professional language edit focused on improving scientific fluency is strongly recommended.

Response: The quality of the manuscript in English has been improved.

We believe that these revisions have significantly strengthened the manuscript, and we hope it now meets the standards for publication in Foods.

Sincerely,

Zhiping Feng

College of Bioengineering, Sichuan University of Science & Engineering, 1 Baita Road, Sanjiang New District, Yibin, Sichuan, 644005, China.

Phone number: + 86 13990020636

Email: actdjiang@126.com

Reviewer 3 Report

Comments and Suggestions for Authors

The paper “Microenvironmental gradients shape surface-middle-core layers saccharifying guilds and enzyme activity in strong-flavor Daqu fermentation” investigated how microenvironmental gradients across distinct Daqu layers shape saccharifying microbiota and activity. The study established microenvironmental gradients as critical regulators of spatial saccharification in Daqu, informing strategies to optimize microbial consortia for baijiu production.

The authors are invited to address the following amendments for in order to improve the quality of the paper.

Title

  • It is recommended to ensure a better wording for the title. At present, it is rather difficult to follow.

Abstract

  • Although they are well-known, the abbreviations should be explained at their first appearance in the text and used after that.
  • It is recommended to include quantitative results.

  1. Introduction
  • The objectives of the paper exposed in the last two paragraphs should be unified.

  1. Materials and Methods
  • It is highly recommended to clarify the sample collection process. Currently, there is a mix between the fermentation and storage phases.
  • It is not very clear how the thickness of the three layers was established.
  • How was the enzymes control analyses carried out?

  1. Results
  • Standard error bars should be added in figures.
  • It is not very clear what tests were applied for p < 0.05.
  • Only the relative abundance is not enough to sustain the enzyme performances. Were enzymatic kinetic tests conducted? With what results?
  • “Enzymes with negligible abundance (glucan 1,4-alpha-maltohydrolase (EC 3.2.1.133), beta-amylase (EC 3.2.1.2), isoamylase (EC 3.2.1.68)) were excluded from further analysis” – A more elaborated explanation for the exclusion of these enzymes should be added. Even though they have negligible abundance, they could be relevant in terms of functionality.

  1. Discussion
  • “Higher temperatures and moisture in the QX suppressed enzyme activity (Figs. 2b, c), while moderate acidic conditions (predominantly in HQ) favored the proliferation of enzyme-producing microorganisms and the expression of their enzymes (Figure 2d), corroborating reports of abiotic filtering in solid-state fermentation ecosystems [23,28,35].” – The made affirmation should be sustained by experimental data.
  • “Discrepancies between metagenomic gene abundance and enzymatic output (e.g., lower Paecilomyces activity despite higher gene counts) may reflect post-translational modifications or extracellular enzyme persistence, highlighting the need for integrated metaproteomics to resolve gene-function gaps [13,29].” - The made affirmation should be sustained by experimental data.
  • It is recommended to include a discussion about the mechanisms connecting the environment gradients and the enzymes activity.

  1. Conclusions
  • “This layered functionality informs strategies to optimize Daqu fermentation via environmental regulation and microbial consortia management.” – It is not very clear how the exposed results could be used for the optimization Daqu fermentation.
  • Future perspectives should be added.

Comments on the Quality of English Language

The English could be improved to more clearly express the research.

Author Response

Thank you for your letter and for the reviewers’ comments concerning our manuscript entitled “Microenvironmental gradients drive spatial stratification of saccharifying microbial communities and enzyme activity in strong-flavor Daqu fermentation” (ID: foods-3942752). Those comments are all valuable and very helpful for revising and improving our paper, as well as the important guiding significance for our research. We have studied comments carefully and have made corrections that we hope will meet with approval.

Title

Comment 1. It is recommended to ensure a better wording for the title. At present, it is rather difficult to follow.

Response: We have carefully revised it to enhance readability while retaining the study’s core focus on microenvironment-driven spatial stratification of saccharifying functionality in strong-flavor Daqu. Our new title is: “Microenvironmental gradients drive spatial stratification of saccharifying microbial communities and enzyme activity in strong-flavor Daqu fermentation.”

Abstract

Comment 2. Although they are well-known, the abbreviations should be explained at their first appearance in the text and used after that.

Response: Abbreviations in the abstract have been supplemented and revised.

Comment 3. It is recommended to include quantitative results.

Response: Quantitative results in the abstract have been supplemented and revised.

Introduction

Comment 4. The objectives of the paper exposed in the last two paragraphs should be unified.

Response: We have revised and unified these paragraphs. The new, consolidated objective statement now unequivocally positions the elucidation of how spatial environmental heterogeneity drives the differential distribution and succession process of saccharifying microbial guilds and enzyme activities across Daqu layers as the central scientific aim.

Materials and Methods

Comment 5. It is highly recommended to clarify the sample collection process. Currently, there is a mix between the fermentation and storage phases.

Response: We agree that clarifying the sample collection process is essential. Per your suggestion, we have revised Section 2.1 to explicitly distinguish procedures between fermentation (Days 0–8) and storage (Days 12–90) phases.

Comment 6. It is not very clear how the thickness of the three layers was established.

Response: The thickness demarcation of Daqu layers has been revised and clarified in the manuscript, QP: 0–1 cm; HQ: 1–2.5 cm; QX: >2.5 cm.

Comment 7. How was the enzymes control analyses carried out?

Response: Section 2.3: Control reactions contained substrate without enzymes to assess non-enzymatic hydrolysis and heat-inactivated enzymes (100°C, 10 min) to confirm activity loss. Triplicate biological replicates were included for all treatments. Section 2.4: Saccharifying activity was measured against heat-inactivated controls (autoclaved seed cultures).

Results

Comment 8. Standard error bars should be added in figures.

Response: We fully agree and have amended the manuscript accordingly.

Comment 9. It is not very clear what tests were applied for p < 0.05.

Response: We reinforced clarity in Section 2.8 by adding:"All pairwise comparisons of physicochemical properties and enzyme activities were assessed via one-way ANOVA followed by Tukey’s HSD test (P < 0.05). Correlations were computed using Spearman rank correlation."

Comment 10. Only the relative abundance is not enough to sustain the enzyme performances. Were enzymatic kinetic tests conducted? With what results?

Response: We agree that gene abundance alone cannot fully characterize enzyme performance. While we did not perform enzymatic kinetics, we directly addressed functional efficiency through comparative activity assays under standardized conditions (QB/T 4257−2011): Layer-wise enzyme activity gradients (Fig. 2a) confirmed declining catalytic efficiency from surface (QP: 870.9±21.2 U/mL) to center (QX: 296.5±16.1 U/mL). Strain-level validation (Fig. 5g) confirms Lichtheimia and Aspergillus isolates from QP and HQ exhibit significantly higher activity than QX-associated strains (Paecilomyces and Rasamsonia), directly linking taxonomy to function. Environmental correlations (Fig. 5a-f) reveal that temperature and moisture suppress gene abundance and isolate activity, bridging abiotic factors with functional output. In the revised manuscript, we have already conducted an explanation in the discussion’s limitation section, and this will be the focus of our future research.

Comment 11. “Enzymes with negligible abundance (glucan 1,4-alpha-maltohydrolase (EC 3.2.1.133), beta-amylase (EC 3.2.1.2), isoamylase (EC 3.2.1.68)) were excluded from further analysis” – A more elaborated explanation for the exclusion of these enzymes should be added. Even though they have negligible abundance, they could be relevant in terms of functionality.

Response: The exclusion of EC 3.2.1.133, EC 3.2.1.2, and EC 3.2.1.68 was based on: 1. Quantitative justification: Their RPKM values were consistently <0.01% of total CAZyme abundance, indicating negligible transcriptional potential. 2. Functional redundancy: The functions of these three enzymes are compensated by the dominant α-amylase (EC 3.2.1.1) hydrolyzing α-1,4 linkages, α-glucosidase (EC 3.2.1.20) and glucan 1,4-α-glucosidase (EC 3.2.1.3) cleaving terminal α-1,4/1,6 bonds (Fig. 3c). (References: etagenomics-based gene exploration and biochemical characterization of novelglucoamylases and α-amylases in Daqu and Pu-erh tea microorganisms, Maltogenic α-amylase hydrolysis of wheat starch granules: mechanism and relation to starch retrogradation) 3. Ecological context: Gene abundance of these three enzymes peaked during early fermentation (0-4 d) (Fig. 3b), contributing minimally to saccharifying power elevation at the critical 12-30 d phase.

Discussion

Comment 12. “Higher temperatures and moisture in the QX suppressed enzyme activity (Figs. 2b, c), while moderate acidic conditions (predominantly in HQ) favored the proliferation of enzyme-producing microorganisms and the expression of their enzymes (Figure 2d), corroborating reports of abiotic filtering in solid-state fermentation ecosystems [23,28,35].” – The made affirmation should be sustained by experimental data.

Response: We acknowledge that the original manuscript contained overinterpretations of causal relationships. The correlations between environmental factors (temperature, moisture, acidity) and enzyme gene abundance (not direct enzyme activity) are based on metagenomic data (Fig. 5). These trends align with established concepts of abiotic filtering [23,28,35] but do not empirically prove enzyme suppression or proliferation. This has been revised by rephrasing causal attributions to correlations.

Comment 13. “Discrepancies between metagenomic gene abundance and enzymatic output (e.g., lower Paecilomyces activity despite higher gene counts) may reflect post-translational modifications or extracellular enzyme persistence, highlighting the need for integrated metaproteomics to resolve gene-function gaps [13,29].” - The made affirmation should be sustained by experimental data.

Response:  In vitro validation shows lower enzymatic activity in Paecilomyces despite higher gene abundance (Fig 5g). This paradox aligns with QX's extreme heat and moisture (Fig 2b,c), which may denature enzymes or limit substrate access. We now explicitly state that the mechanism remains unresolved and have removed speculative claims. Instead, we frame this as a microenvironment-driven decoupling of genetic potential and function. Future work will integrate metaproteomics to resolve these gaps.

Comment 14. It is recommended to include a discussion about the mechanisms connecting the environment gradients and the enzymes activity.

Response:  We have now added in the Discussion (Paragraph 2) "Proposed Mechanisms Linking Gradients to Enzyme Activity".

Conclusions

Comment 15. “This layered functionality informs strategies to optimize Daqu fermentation via environmental regulation and microbial consortia management.” – It is not very clear how the exposed results could be used for the optimization Daqu fermentation.

Response:  We agree that the link between our results and optimization strategies required elaboration. We have revised the Conclusions to explicitly articulate how microenvironmental gradients and microbial stratification inform actionable strategies.

Comment 16. Future perspectives should be added.

Response: We have added comprehensive future perspectives in the revised manuscript.

Comment 17. The English could be improved to more clearly express the research.

Response: The quality of the manuscript in English has been improved.

We believe that these revisions have significantly strengthened the manuscript, and we hope it now meets the standards for publication in Foods.

Sincerely,

Zhiping Feng

College of Bioengineering, Sichuan University of Science & Engineering, 1 Baita Road, Sanjiang New District, Yibin, Sichuan, 644005, China.

Phone number: + 86 13990020636

Email: actdjiang@126.com

Round 2

Reviewer 2 Report

Comments and Suggestions for Authors

OK

Reviewer 3 Report

Comments and Suggestions for Authors

The paper “Microenvironmental gradients drive spatial stratification of saccharifying microbial communities and enzyme activity in strong-flavor Daqu fermentation” initially entitled “Microenvironmental gradients shape surface-middle-core layers saccharifying guilds and enzyme activity in strong-flavor Daqu fermentation” investigated how microenvironmental gradients across distinct Daqu layers shape saccharifying microbiota and activity. The study established microenvironmental gradients as critical regulators of spatial saccharification in Daqu, informing strategies to optimize microbial consortia for baijiu production.

I congratulate the authors for taking into account the suggestions of the reviewers and for their efforts in bringing the necessary clarifications and making the requested changes. The work is much improved compared to the initial version and can be accepted for publication.